# lncRNA MSTRG4710 Promotes the Proliferation and Differentiation of Preadipocytes through miR-29b-3p/IGF1 Axis

**DOI:** 10.3390/ijms242115715

**Published:** 2023-10-28

**Authors:** Tao Tang, Genglong Jiang, Jiahao Shao, Meigui Wang, Xiaoxiao Zhang, Siqi Xia, Wenqiang Sun, Xianbo Jia, Jie Wang, Songjia Lai

**Affiliations:** 1College of Animal Science and Technology, Sichuan Agricultural University, Chengdu 611130, China; 2Farm Animal Genetic Resources Exploration and Innovation Key Laboratory of Sichuan Province, Sichuan Agricultural University, Chengdu 611130, Chinawjie68@163.com (J.W.)

**Keywords:** rabbits, lncRNA-MSTRG4710, miR-29b-3p, *IGF1*, preadipocytes, cell proliferation and differentiation

## Abstract

Obesity, a major global health issue, is increasingly associated with the integral role of long non-coding RNA (lncRNA) in adipogenesis. Recently, we found that lncRNA-MSTRG4710 was highly expressed in the liver of rabbits fed a high-fat diet, but whether it is involved in lipid metabolism remains unclear. A series of experiments involving CCK-8, EDU, qPCR, and Oil Red O staining demonstrated that the overexpression of MSTRG4710 stimulated the proliferation and differentiation of preadipocytes while its knockdown inhibited these processes. Bioinformatics analysis showed that miR-29b-3p was a potential target gene of MSTRG4710, and IGF1 was a downstream target gene of miR-29b-3p. Luciferase reporter gene analysis and qPCR analysis confirmed that miR-29b-3p was a potential target gene of MSTRG4710, and miR-29b-3p directly targeted the 3′UTR of *IGF1*. The overexpression of miR-29b-3p was observed to regulate IGF1 protein and mRNA levels negatively. Additionally, a total of 414 known differentially expressed genes between the miR-29b-3p mimic, miR-29b-3p negative control (NC), siMSTRG4710, and siMSTRG4710-NC group were screened via transcriptome sequencing technology. The GO- and KEGG-enriched pathways were found to be related to lipid metabolism. The study also established that miR-29b-3p targets IGF1 to inhibit preadipocyte proliferation and differentiation. Notably, *IGF1* knockdown significantly reduced preadipocyte proliferation and differentiation. Furthermore, co-transfection of pcDNA3.1(+)-MSTRG4710 and mimics into rabbit preadipocytes revealed that the mimics reversed the promotional effect of pcDNA3.1(+)-MSTRG4710. In conclusion, these results uncover that MSTRG4710 positively regulated cell proliferation and adipogenesis by the miR-29b-3p/*IGF1* axis. Our findings might provide a new target for studying adipogenesis in rabbit preadipocytes and obesity.

## 1. Introduction

Obesity, a metabolic syndrome engendered by an imbalance between energy intake and expenditure, elevates the risk of numerous diseases, including diabetes, atherosclerosis, and cardiovascular disease [1]. Reports indicate that by the end of 2016, 39% of adults were overweight and 18% were obese. Forecasts show that, by 2030, more than 2.16 billion individuals will be overweight and 1.12 billion obese [2]. Obesity can also lead to psychological problems such as depression, anxiety, and low self-esteem [3]. Considering the apparent threat of obesity to public health and the economy, the World Health Organization has proposed a goal to arrest the rise of obesity by 2025 [2]. The occurrence of obesity is attributed to a multitude of factors, with the primary link being the excessive proliferation and differentiation of adipocytes, resulting in an overproduction of adipose cells [4]. However, adipose tissue plays a pivotal role in body health regulation. It modulates glucose and lipid metabolism throughout the body by secreting a variety of adipokines, such as leptin, adiponectin, palmitoleate, and fatty acid esters of hydroxy fatty acids [5].

Long non-coding RNAs (lncRNAs), defined as RNAs exceeding 200 nucleotides in length without protein-encoding capacity [6], were previously thought to lack biological function due to their non-involvement in protein coding, thus considered transcriptional noise [7]. Until recent years, a flow of research had suggested that various lncRNAs played critical roles in the biological process, such as in cell proliferation, differentiation, apoptosis, and tissue and organ development [8,9]. LncRNAs exist in the nucleus, cytoplasm, and organelles. Still, the number of lncRNAs in the nucleus is higher than that in the cytoplasm and organelles, and the lncRNAs in different parts play different functions. Studies have shown that lncLSTR, which is a liver-specific triglyceride-regulated RNA, is a lncRNA that can be expressed explicitly in mouse liver and regulates the energy metabolism and lipid metabolism in the liver by directly controlling Cyp8b1, one of the rate-limiting enzymes responsible for bile acid synthesis [10]. LncBATE10 can promote the differentiation of brown adipocytes and the effect of fat browning by competitively binding to Celf1, which is a tan RNA-binding protein, to induce Pgc1a release [11].

microRNAs (miRNAs) are small non-coding RNA molecules of 21 to 23 nucleotides in length, which are categorized as a class of highly conserved endogenous short non-coding RNAs [12]. miRNAs exist ubiquitously in all eukaryotes, serving as a post-transcriptional regulatory factor of the protein-encoding mRNA. It mainly acts by base-pairing with the complementary sequence of the 3’UTR region of the target gene, thereby promoting the degradation of mRNA and/or inhibiting its translation [13]. miRNAs are mRNA-inhibitory factors that regulate various physiological and cellular processes, including fat formation, cell proliferation, apoptosis, triglyceride synthesis, and lipolysis, via post-transcriptional processing [14]. In recent years, there has been increasing evidence that lncRNAs can regulate genes by directly binding the mRNA competing with miRNAs, thus regulating physiological processes, similar to endogenous RNA (ceRNAs) [15]. LncADNCR is the most downregulated lncRNA among the differentially expressed lncRNAs in bovine preadipocytes and differentiated adipocytes. Its interaction with miR-204 leads to increased expression of SIRT1, which is a target gene of miR-204. This, in turn, inhibits adipocyte differentiation. The underlying mechanism involves SIRT1 binding with NCOR and SMART, thereby impeding the activity of PPARγ and suppressing adipogenic gene expression and adipocyte differentiation [16].

Our previous sequencing data revealed that an uncharacterized lncRNA-MSTRG4710 exhibited high expression levels in the liver of obese rabbits. Considering the liver’s role as a site for de novo fat synthesis, this finding strongly suggests that MSTRG4710 likely plays a regulatory role in adipogenesis. In addition, MSTRG4710 is also highly expressed in the adipose tissue of obese rabbits, and there is a high expression of miR-29b-3p related to MSTRG4710. However, the specific regulation mechanism of lipid metabolism is still unclear [17,18]. In this study, we found that MSTRG4710 acted as a molecular sponge for miR-29b-3p to inhibit miR-29b-3p expression and promote *IGF1* expression, which, in turn, enabled the proliferation and differentiation of rabbit preadipocytes and could serve as a potential therapeutic target for obesity treatment.

## 2. Results

### 2.1. Tissue Expression Profiles of LncRNA-MSTRG4710 and miR-29b-3p in Rabbits

To investigate the expression of miR-29b-3p and MSTRG4710 in different rabbit tissues, lung, heart, muscle, perirenal fat, liver, spleen and kidney tissues from 30-day-old female Tianfu Black rabbits were selected, and the tissue expression profile of miR-29b-3p was determined via qPCR (Figure 1A,B). Based on the expression profile analysis, both miR-29b-3p and MSTRG4710 were found to be expressed in multiple tissues throughout the body, with the lowest expression observed in muscle. The expression levels of miR-29b-3p and MSTRG4710 were found to be higher in perirenal adipose tissues compared to other body tissues. These findings suggest a potential association between lncRNA-MSTRG4710, miR-29b-3p, and fat development.

### 2.2. Isolation and Culture and Establishment of Rabbit Differentiation Model of Rabbit Preadipocytes

To establish the differentiation model of rabbit preadipocytes, the preadipocytes were induced to differentiate using the method mentioned above when their density reached 80%. The Oil Red O staining results demonstrated an increase in the number of lipid droplets during the progression of differentiation (Figure 2A). Moreover, an enzyme label was employed to measure the absorbance at 510 nm to further verify the induction of differentiation in preadipocytes. As illustrated in Figure 2B, the OD value demonstrated a consistent upward trend during induced differentiation despite a slight decrease on day 6. Notably, the OD value remained significantly higher than the pre-differentiation level. Concurrently, the expression of critical genes involved in preadipocyte differentiation, namely *PPARγ* and *C/EBPα*, was observed to be upregulated in the differentiation model. Specifically, the expression of *PPARγ* reached its peak on the 3rd day, while *C/EBPα* exhibited its highest expression level on the 4th day (Figure 2C,D).

### 2.3. Effect of LncRNA-MSTRG4710 on Proliferation and Differentiation of Rabbit Preadipocytes

#### 2.3.1. LncRNA-MSTRG4710 Promotes the Proliferation of Rabbit Preadipocytes

To ascertain the complete sequence of both ends of the lncRNA-MSTRG4710, we conducted 5′ and 3′ RACE assays specifically in the preadipocytes of rabbits (Appendix A). The entire sequence of lncRNA-MSTRG4710 was successfully constructed on the pcDNA3.1(+) vector. To investigate the effect of MSTRG4710 on the proliferation of rabbit preadipocytes, subsequent experiments, including CCK-8, EDU, qPCR, and Western Blot (WB), were performed. The qPCR results indicated a significant decrease in the expression of MSTRG4710 in the siMSTRG4710 group, while the pcDNA3.1(+)-MSTRG4710 group showed a significant increase, thus confirming the successful transfection of lncRNA (Figure 3A). On the second day of proliferation, the EDU proliferation experiments revealed a significantly higher positive cell rate in the pcDNA3.1(+)-MSTRG4710 group compared to the pcDNA3.1(+) group. Additionally, the siMSTRG4710 group exhibited the opposite trend (Figure 3C,D). The CCK-8 results demonstrated a significant increase in absorbance at 0 h, 24 h, 48 h, and 72 h after transfection in the pcDNA3.1(+)-MSTRG4710 group. Conversely, the siMSTRG4710 group exhibited the opposite trend with a decrease in absorbance (Figure 3F,G). Furthermore, on the second day of proliferation, we examined the expression of proliferation-related genes at protein and mRNA levels. WB and qPCR results had the same trend (Figure 3E,H). The findings revealed that lncRNA-MSTRG4710 exerts a promotional effect on the proliferation of rabbit preadipocytes.

#### 2.3.2. LncRNA-MSTRG4710 Promotes the Differentiation of Rabbit Preadipocytes

To investigate the function of MSTRG4710 in the differentiation process of rabbit preadipocytes, we examined the expression of MSTRG4710 MSTRG4710 expression over 8d. The results revealed that the expression of MSTRG4710 peaked at 4d in preadipocytes (Figure 4A). We transfected siMSTRG4710 and pcDNA3.1(+)-MSTRG4710 into rabbit preadipocytes, and two days after induction of differentiation, we examined the expression of MSTRG4710, which was confirmed by the results shown in Figure 4B. The Oil Red O staining, qPCR, and WB were used to identify the function of MSTRG4710 on the differentiation of rabbit preadipocytes. The Oil Red O staining results demonstrated a significant increase in lipid droplets in the pcDNA3.1(+)-MSTRG4710 group after 2 days of transfection. Conversely, the interference group exhibited the opposite trend, with decreased lipid droplets (Figure 4C,D). The qPCR results showed an apparent and highly significant increase in genes in the pcDNA3.1(+)-MSTRG4710 group, including *PPARγ*, *FABP4*, *SREBP1*, and *C/EBPα* in contrast, these genes were expressed at lower levels in the inhibitor group (Figure 4E). The PPARγ and FABP4 protein expression levels showed the same results (Figure 4F,G). These results indicate that MSTRG4710 promotes the differentiation of rabbit preadipocytes.

### 2.4. Effect of miR-29b-3p Proliferation and Differentiation of Rabbit Preadipocytes

#### 2.4.1. miR-29b-3p Inhibited the Proliferation of Rabbit Preadipocytes

Adipocytes, as the primary constituents of fat tissue, play a vital role in accumulating adipose mass. Preadipocyte proliferation and subsequent differentiation are essential for substantially increasing adipose tissue mass [19,20]. Thus, adipocyte proliferation and differentiation are essential components of lipogenesis. We studied the effect of miR-29b-3p on the proliferation of preadipocytes. After preadipocytes were transfected for 48 h with NC, mimics, inhibitor NC, or inhibitor, we detected the genes *CDK2* and *CDK3* positively related to cell proliferation by qPCR. The results are shown in Figure 5A,B. The overexpression of miR-29b-3p resulted in a significant decrease in the expression levels of *CDK2* and *CDK3* genes, while the inhibitor group exhibited the opposite trend. Furthermore, we conducted EDU incorporation experiments to evaluate the effect of miR-29b-3p on proliferation. As shown in Figure 5C,D, the miR-29b-3p mimics reduced the proliferation of preadipocytes. In contrast, the miR-29b-3p inhibitor significantly increased preadipocyte proliferation. We conducted a CCK-8 experiment to verify further the effect of miR-29b-3p on the proliferation of preadipocytes. Similarly, the CCK-8 assay showed that the cell proliferation rate reduced significantly in preadipocytes transfected with the miR-29b-3p mimics but increased substantially in response to the inhibitor (Figure 5E,F). Therefore, our data indicate that miR-29b-3p inhibits preadipocyte proliferation.

#### 2.4.2. miR-29b-3p Inhibits the Differentiation of Rabbit Preadipocytes

To investigate the function of miR-29b-3p in the differentiation of rabbit preadipocytes, we examined the expression pattern of miR-29b-3p in preadipocytes during an 8 d period. Interestingly, we observed that the expression of miR-29b-3p reached its highest level on day 2 of the differentiation process (Figure 6B). Then, mimics, NC, inhibitor and inhibitor NC were transfected into preadipocytes, and the expression of miR-29b-3p in the inhibitor group was significantly decreased, as shown in Figure 6D, and that in the mimics group was increased considerably. The Oil Red O staining, qPCR, and WB were used to identify the function of miR-29b-3p on the differentiation of rabbit preadipocytes. The Oil Red O staining showed that the number of lipid droplets was lower in the mimics group than in the NC group. Still, lipid droplets were higher in the inhibitor group than in the inhibitor NC group (Figure 6A,C). At 2 d, 4 d, and 6 d post-transfection, we observed a significant decrease in the expression of *C/EBPα* in the mimics group, while the inhibitor group exhibited an increase in its expression. Similarly, the expression of *PPAR*γ was notably reduced in the mimics group but increased in the inhibitor group at 2 d after transfection (Figure 6E,F). The protein level of PPARγ was significantly decreased in the mimics group but increased in the inhibitor group, but the difference was insignificant. (Figure 6G,H). These results indicate that miR-29b-3p promotes the differentiation of rabbit preadipocytes.

### 2.5. miR-29b-3p Inhibits the Proliferation and Differentiation of Rabbit Preadipocytes by Targeting IGF1

In order to investigate the potential function of miR-29b-3p, we utilized TargetScan 7.2 and miRWalk databases to predict its target genes. Subsequently, GO and KEGG analyses were performed on the predicted target genes. Interestingly, the results revealed that these target genes were implicated in critical pathways such as the AMPK and insulin signaling pathways. (Appendix A). Cytoscape_v3.6.1 software was used to draw the relationship diagram based on KEGG data (Appendix A) to explore the relationship between target genes and their pathways. We found that *IGF1* is involved in multiple pathways related to energy metabolism and adipogenesis. Subsequently, TargetScan was used to predict target gene IGF1 and miR-29b-3p binding sites. It was found that there was a complementary binding site of eight bases in the 3′UTR region of miR-29b-3p and *IGF1*, and its binding site was conserved in many species, such as human, cow, dog, mouse, etc. (Appendix A). Based on the initial prediction, *IGF1* was identified as a potential target gene of miR-29b-3p. This led us to select *IGF1* for further experimental investigation.

To further verify the relationship between miR-29b-3p and *IGF1*, we examined the mRNA expression levels of *IGF1* in both the mimics and inhibitor groups throughout the differentiation process. Our results demonstrated a significant decrease in the expression of *IGF1* in the mimics group at 2d, 4d, and 6d. In contrast, in the inhibitor group, there was a significant increase in its expression at 2d, 6d, and 8d (Figure 7A). The expression of miR-29b-3p is shown in Figure 7B. Furthermore, we observed a significant decrease in the expression of *IGF1* in the si-IGF1 group during both the proliferation and differentiation stages of preadipocytes. Notably, the expression of *IGF1* in the mimics + si-IGF1 group was even lower than that in the si-IGF1 group. On the contrary, the inhibitor group exhibited a significantly higher expression of *IGF1* compared to the inhibitor + si-IGF1 group (Figure 7C,D). In addition, we observed a significant decrease in the protein expression of IGF1 in the mimics group, while it was upregulated in the inhibitor group (Figure 7O,R). The above results proved the targeting relationship between miR-29b-3p and *IGF1*, which was also confirmed by dual luciferase reporter assay (Figure 7E,F).

The EDU proliferation assay demonstrated a significant increase in positive cells in the si-IGF1-NC group compared to the si-IGF1 group (Figure 7G,H). The CCK-8 results showed a significant decrease in absorbance in the si-IGF1 and mimics + si-IGF1 group at 0 h, 24 h, 48 h and 72 h after transfection, while the inhibitor + si-IGF1 group showed the opposite trend (Figure 7I). The expression levels of *CDK2*, *CDK3*, *CDK4*, and *PCNA* in the si-IGF1 and mimics+si-IGF1 groups were significantly decreased, while those in the inhibitor + si-IGF1 group were increased considerably. WB also showed the same results (Figure 7J,Q,T). The expression of *IGF1* exhibited a gradual increase throughout preadipocyte differentiation, reaching its peak on 4d. These findings suggest a potential association between *IGF1* and the differentiation of preadipocytes (Figure 7K). The Oil Red O staining showed that lipid droplets significantly increased in the inhibitor + si-IGF1 group. Still, the number of lipid droplets decreased considerably in the si-IGF1 and mimics + si-IGF1 groups after 2 d of transfection. Moreover, compared with the si-IGF1 group, the mimics + si-IGF1 group decreased more (Figure 7L,M). The qPCR and WB also showed the same results (Figure 7N,P,S). Based on the results mentioned above, it can be inferred that miR-29b-3p plays a crucial role in inhibiting the proliferation and differentiation of rabbit preadipocytes by targeting *IGF1*.

### 2.6. MSTRG4710 Promotes the Differentiation of Rabbit Preadipocytes via the miR-29b-3p/IGF1 Pathway

To verify the relationship between MSTRG4710 and miR-29b-3p. The qPCR was used to detect the expression levels of miR-29b-3p, MSTRG4710, and *IGF1* under various treatment conditions during the rabbit differentiation of preadipocytes. In the siMSTRG4710 group, there was a significant increase in the expression of miR-29-3p. Conversely, in the pcDNA3.1(+) MSTRG4710 group, there was a significant decrease in the expression of miR-29-3p (Figure 8A). Meanwhile, the expression of miR-29-3p in the mimics + siMSTRG4710 group was significantly increased, while that in the inhibitor + siMSTRG4710 and inhibitor + pcDNA3.1(+)-MSTG4710 group was significantly decreased (Figure 8E). The expression of MSTRG4710 in the mimics, mimics + siMSTRG4710, mimics + pcDNA3.1(+)-MSTG4710, mimics + si-IGF1 and si-IGF1 group was significantly decreased, while that in the inhibitor, inhibitor + si-IGF1 and inhibitor + pcDNA3.1(+)-MSTG4710 group was significantly reduced (Figure 8C,D). The expression level of *IGF1* was opposite to that of miR-29b-3p (Figure 8B,F). The dual luciferase reporter assay confirmed the targeting relationship between MSTRG4710 and miR-29b-3p (Figure 8G,H). Finally, WB and qPCR were used to verify whether MSTGR4710 affected preadipocyte differentiation through the miR-29b-3p/IGF1 pathway. The results indicate that the downregulation of *IGF1* expression inhibited differentiation, while the upregulation of *IGF1* expression promoted differentiation (Figure 8I–K). Our results suggested that MSTRG4710 could promote rabbit preadipocyte differentiation by inhibiting miR-29b-3p expression and further promoting *IGF1* expression by acting as a molecular sponge for miR-29b-3p.

### 2.7. MSTRG4710 Promotes the Proliferation of Rabbit Preadipocytes via the miR-29b-3p/IGF1 Pathway

During the proliferation phase of rabbit preadipocytes, the expression of miR-29b-3p showed a significant increase in the siMSTRG4710 group, while it exhibited a significant decrease in the pcDNA3.1(+)-MSTRG4710 group (Figure 9A). The expression of MSTRG4710 was significantly decreased in mimics, mimics + siMSTRG4710, si-IGF1 and mimics + si-IGF1 groups. Furthermore, the decrease was more significant in the mimics + siMSTRG4710 group than in the mimics group and more in the si-IGF1 group than in the mimics + si-IGF1 group. The expression of MSTRG4710 was significantly increased in inhibitor, mimics + pcDNA3.1(+)-MSTG4710, inhibitor+pcDNA3.1(+)-MSTG4710 si-IGF1 and inhibitor + si-IGF1 groups. Moreover, the inhibitor+pcDNA3.1(+)-MSTG4710 group decreased more than the mimics + pcDNA3.1(+)-MSTG4710 group (Figure 9B,C). We then examined the amount of *IGF1* expression. The knockdown of MSTRG4710 decreased the expression of *IGF1*, and the decrease was more significant in the mimics+siMSTRG4710 group. The overexpression of MSTRG4710 increased the expression of *IGF1*, while the addition of mimics decreased the expression of *IGF1*. Furthermore, the expression of *IGF1* was upregulated in the inhibitor+siMSTRG4710 group (Figure 9E). The expression levels of *PCNA*, *CDK2*, *CDK3* and *CDK4* were significantly downregulated in the mimics+siMSTRG4710 group and significantly downregulated in the inhibitor+siMSTRG4710 group. Mimics/inhibitor + pcDNA3.1(+)-MSTG4710 group was significantly upregulated, which was consistent with the expression of *IGF1* (Figure 9G). The WB and CCK-8 proliferation assays were consistent with the above results (Figure 9D,F,H). Therefore, our findings lead us to conclude that MSTRG4710 functions as a molecular sponge for miR-29b-3p, effectively suppressing its expression and consequently promoting the expression of its downstream target gene *IGF1.* This regulatory mechanism ultimately facilitates the proliferation of rabbit preadipocytes.

### 2.8. Analysis of Differentially Expressed Genes

To further investigate the functions of miR-29b-3p and MSTRG4710, the miR-29b-3p mimic group (mimics), NC group (NC), siMSTRG4710 group and siMSTRG4710-NC group were sent to the company for transcriptome sequencing. As shown in Table 1, a total of 683,014,766 clean readings were obtained. The level of Q30 in mimics, NC, siMSTRG4710 and siMSTRG4710-NC groups was equivalent (percentage of readings with Phred quality value > 30), ranging from 94.12% to 95.47%. The GC content of the libraries ranged from 56.41% to 57.32%, with an average content of 56.70%. The comparative analysis showed that the average comparison rate of samples was 87.89%. The sequence number and percentage of all samples aligned to exon, intron and intergenic region were 67.40%, 15.69% and 16.90%, respectively. Therefore, all libraries were of high quality and could be used for further analysis. In the mimics and NC group, 279 (124 upregulated and 155 downregulated) known differentially expressed genes were screened from the sequencing library using DESeq2. Values of log2 (fold change) and -log10 (*p*-value) were used to construct volcano figures for differentially expressed genes (Figure 10A). Similarly, 135 differentially expressed genes were detected between siMSTRG4710 and siMSTRG4710-NC groups, of which 40 genes were upregulated and 95 genes were downregulated (Figure 10B). Clustering heat showed that the functionally related genes in each group had similar expression patterns under the same conditions. Still, the expression levels were different among the groups, indicating that the samples of each group were separated strongly (Figure 10C,D). The differentially expressed genes were analyzed and annotated using NCBI, Uniprot, GO and KEGG databases to obtain detailed descriptions of differentially expressed genes. Finally, in the mimics and NC group, the GO analysis results showed that 1655 GO items were enriched (1141 biological processes (BP), 223 cell composition (CC), and 291 molecular functions (MF)) (Figure 10E). In the siMSTRG4710 and siMSTRG4710-NC groups, the GO analysis results showed that 748 GO items were enriched (485 BP, 127 CC, and 136 MF) (Figure 10F). The main biological processes involved cell population proliferation, metabolic function and growth. KEGG analysis showed that a total of 261 pathways were enriched, among which the PI3K-AKT signaling pathway, Wnt signaling pathway, NOD-like receptor signaling pathway, TNF signaling pathway, JAK-STAT signaling pathway and cell cycle were significantly enriched (Figure 10G,H).

## 3. Discussion

Adipose tissue is an endocrine and energy-saving organ [21]. The primary function of adipose tissue is to store energy and maintain the body’s metabolic balance [22]. Adipocytes, mesenchymal cells, are crucial in storing surplus dietary energy as fat within lipid droplets. However, excessive energy is stored as triglycerides in cases of over-nutrition, resulting in an imbalance between energy intake and expenditure. This leads to rapid adipose tissue expansion, disrupting lipid homeostasis and lipogenesis [23,24]. When the normal function of adipocytes is disrupted, it can result in metabolic disorders, such as obesity and type 2 diabetes [25]. Adipose tissue formation is a complex biological process that involves the differentiation of preadipocytes into mature adipocytes as a crucial step. In vitro, the differentiation of adipocytes requires the addition of exogenous inducers [26]. In addition, *C/EBPα*, *PPARγ*, *SREBP1* and *FABP4* activation is essential for adipogenesis and adipocyte differentiation [27,28,29,30]. In line with previous research, our findings confirm that the number and size of lipid droplets increase during preadipocyte differentiation. Moreover, we observed a positive correlation between the expression of *C/EBPα* and *PPARγ* and the process of adipogenesis [31].

LncRNAs are increasingly recognized as crucial regulators that impact diverse biological processes and contribute to the pathogenesis of obesity [32,33]. Recent studies have found that lncRNA-mLas-V3 may play a protective role against stress associated with adipogenesis, and its absence has been linked to apoptosis [34]. In this study, we selected an unannotated lncRNA-MSTRG4710, which exhibited high expression levels in the liver tissue of obese rabbits. Considering that the liver is a significant site for de novo fat synthesis, we hypothesized that MSTRG4710 might be associated with lipid metabolism [35]. Then, we examined the expression of MSTRG4710 in various tissues of Tianfu black rabbits at 35 and 70 days of age and found that MSTRG4710 was highly expressed in perirenal adipose tissues except the liver. These findings indicate a close association between this lncRNA-MSTRG4710 and adipose tissue development. LncRNAs display more muscular tissue-specific expression patterns, suggesting integral roles in cell type-specific processes [36,37]. Therefore, we hypothesized that MSTRG4710 has a regulatory effect on adipose development in rabbits. Numerous studies have linked various lncRNAs to preadipocyte differentiation and adipogenesis, highlighting the role of altered lncRNA expression in the development and progression of obesity [38,39]. The lncMYOZ2 mediates an AHCY/MYOZ2 axis to promote adipogenic differentiation in porcine preadipocytes [40]. Our study also revealed that the overexpression of MSTRG4710 upregulated *PPARγ*, *C/EBPα* gene expression, along with an increase in the number of lipid droplets. Conversely, the inhibition of MSTRG4710 inhibited the differentiation process of preadipocytes, indicating that MSTRG4710 could promote rabbit preadipocyte differentiation. Meanwhile, preadipocyte proliferation is one factor in determining fat deposition, and studies have shown that lncRNA has a regulatory effect on the proliferation of preadipocytes [41,42]. The *PCNA* and *CDK* gene family is thought to regulate cell proliferation [43,44]. In this study, we concluded that overexpressing and repressing the expression of MSTRG4710 could promote rabbit preadipocyte proliferation.

miRNAs are ubiquitous in both prokaryotic and eukaryotic organisms, present in diverse tissues, organs and species, thereby affecting various metabolism in the body. Studies have found that miR-26a modulates insulin-induced adipogenic differentiation of ADSCs through the CDK5/FOXC2 pathway [45]. miR-148a-3p has been shown to modulate the differentiation of rabbit preadipocytes by downregulating PTEN expression and promoting TG accumulation in adipocytes [26]. miR-29b-3p is involved in the occurrence of cardiovascular diseases and tumors [46,47]. In addition, the lncRNA-MIR99AHG sponges miR-29b-3p to promote PPARγ-mediated regulation of preadipocyte differentiation, which may be involved in obesity [48]. In our previous study, we observed a significant upregulation of miR-29b-3p expression in the adipose tissue of rabbits subjected to a high-fat diet [17]. In this study, the expression of miR-29b-3p in different tissues of 35-day-old rabbits was detected, and it was found that the highest in the liver, and it also had increased expression in perirenal adipose tissue, indicating that this miRNA has a specific role in lipogenesis. At the stage of adipocyte differentiation, miR-29b-3p plays a negative role in preadipocyte differentiation. The upregulation of miR-29b-3p decreased the adipogenic marker genes *PPARγ* and *C/EBPα* expression. In contrast, using a synthetic inhibitor to downregulate miR-29b-3p significantly enhanced neutral lipid droplet formation and stimulated the expression of marker genes at the PPARγ protein level. At the stage of adipocyte proliferation, to visually determine the functional consequences of miR-29b-3p in preadipocyte proliferation, we performed EDU and CCK assays, which showed that the number of EDU-positive cells and the absorbance of preadipocytes were significantly reduced in the miR-29b-3p mimic group. In addition, the expression levels of proliferation-related genes *CDK2* and *CDK3* were significantly decreased in the mimics group. Taken together, these findings suggest that miR-29b-3p may exert a negative regulatory effect on preadipocyte proliferation and differentiation.

miRNAs serve multiple functions, one of which is endogenous non-coding RNAs that participate in post-transcriptional expression regulation by specifically binding to the 3′UTR of downstream target genes and inhibiting the translation of mRNA [49]. In this study, a total of 563 target genes were identified using the online software TargetScan (http://www.targetscan.org/vert_71/, accessed on 18 December 2021) and further analysis was performed using GO and KEGG to obtain an overview of these target genes and further explore the function of miR-29b-3p. We found that the target genes of miR-29b-3p were mainly enriched in the PI3K-AKT signaling pathway, ECM-receptor interaction, type II diabetes, adipokine signaling pathway, AMPK signaling pathway, thyroid hormone signaling pathway, and FoxO signaling pathway. These pathways have been identified as critical to adipocyte differentiation [50,51]. These findings suggest that miR-29b-3p may significantly impact various cellular processes and metabolic pathways related to adipocyte function and metabolism. The PPI network showed that the target gene *IGF-1* was involved in the PI3K-AKT signaling pathway, AMPK signaling pathway and FoxO signaling pathway. Therefore, we speculated that miR-29b-3p may regulate the proliferation and differentiation of rabbit preadipocytes by targeting the *IGF1* gene. Previous studies have shown that *IGF1* can promote the proliferation and differentiation of various cells, including preadipocytes [52]. During the process of human preadipocyte differentiation, there is an upregulation in the expression of *IGF1* and *IGFBP-3*. Additionally, both preadipocytes and adipocytes secrete *IGF1*, *IGF2*, and *IGFBP-3*. As a result, *IGF1* is recognized as one of the critical factors in regulating adipose tissue metabolism [53]. We observed that the mRNA and protein levels of *IGF1* were decreased when miR-29b-3p was overexpressed. After interfering with the expression of *IGF1*, the expression of miR-29b-3p increased. In addition, the dual luciferase system also found that miR-29b-3p had a targeting relationship with *IGF1*. In addition, CCK-8, EDU, Oil Red O and other results found that the proliferation and differentiation of preadipocytes had a positive regulatory relationship with *IGF1*. Based on these results, we propose that miR-29b-3p exerts its inhibitory effects on rabbit preadipocyte proliferation and differentiation by directly targeting *IGF1*.

LncRNAs can function as signals, decoys, guides, and scaffolds, regulating the expression of their target genes at transcriptional and post-transcriptional levels [54]. Research has demonstrated that lncRNAs can act as molecular sponges for miRNAs, preventing miRNAs from binding to mRNA and inhibiting mRNA degradation. For example, lncSEMT regulates skeletal muscle differentiation and increases muscle fiber number by controlling *IGF2* abundance by antagonizing miR-125p in vivo and in vitro [55]. Some lncRNAs can coactivate *PPARγ*. For example, SRA can link the balance of insulin and adipogenesis through SRA-responsive genes in adipocytes and PPARγ-dependent reporter genes [56]. Our findings indicate that MSTRG4710 functions as a miR-29b-3p sponge, effectively suppressing its expression and leading to an upregulation of *IGF1*. Thus, it promotes the proliferation and differentiation of preadipocytes. Our study used RNA-seq technology to investigate the regulatory effects of miR-29b-3p and MSTRG4710 on preadipocytes in rabbits. Moreover, 414 known differentially expressed genes were screened from the sequencing library using DESeq2. Moreover, its target gene, *IGF1*, was significantly downregulated after the overexpression of miR-29b-3p. GO and KEGG enriched the pathways related to lipid metabolism, proliferation, differentiation and cell cycle. For example, the canonical wnt signaling pathway is a central negative regulator of adipogenesis [57]. The activation of Wnt/β-catenin signaling via the overexpression of Wnt1, Wnt6, Wnt10a and Wnt10b prevented the induction of *PPARγ* and *C/EBPα* and inhibited adipogenesis through a β-catenin dependent mechanism [58]. The Jak-STAT pathway serves as a crucial signaling mechanism for various cytokines and growth factors. The activation of Jak proteins stimulates several cellular processes, including cell proliferation, differentiation, migration, and apoptosis. These events are vital for critical biological processes such as hematopoiesis, immune development, mammary gland development and lactation, adipogenesis, sexual growth, and other physiological processes [59,60]. Therefore, miR-29b-3p and MSTRG4710 may be essential regulators in adipogenesis.

## 4. Materials and Methods

### 4.1. Animals and Sample Collection

The Institutional Animal Care and Use Committee of the College of Animal Science and Technology, Sichuan Agricultural University, China, approved all experimental procedures using rabbits in our study. The perirenal adipose tissues were collected from three Tianfu black rabbits, 3 days postnatal, raised under standard conditions at the Sichuan Agricultural University farm in Yaan, Sichuan, China. After collection, all samples were immediately frozen in liquid nitrogen and stored at −80°C until further analysis.

### 4.2. Culture and Differentiation of Rabbit Preadipocytes

Preadipocytes were isolated from the perirenal adipose tissue of 3 d postnatal Tianfu black rabbits under sterile conditions. The tissues were placed in a 6-well cell culture plate containing Phosphate-Buffered Saline (PBS) of 4% penicillin–streptomycin (Gibco) solution. The connective tissue and blood vessels were removed with sterile surgical scissors and forceps. The collected preadipose tissue was washed with PBS, and the treatment was repeated three times. Transfer the excisional tissue to a 15 mL centrifuge tube containing 0.1% type I collagenase (Gibco, Carlsbad, CA, USA), and place it in a 37 °C water bath for digestion for 1 h, shaking every 15 min. Once the tissue was minced, aan equal amount of complete growth medium was added to the centrifuge tube to stop digestion. Then, the mixed culture medium was filtered using a cell sieve of 40 nm and 70 nm, respectively. The preadipocytes were seeded into a T25 culture flask at 37 °C in a humidified incubator with 5% CO_2_, and the media was replaced every 2–3 days. The cells will be passaged when the cell density reaches 70–80%. The cells were then passaged to the third generation and cryopreserved in liquid nitrogen for future studies. Once the cell density reached 80%, they were cultured in a differentiation medium consisting of 5% FBS, 1 μmol/L dexamethasone (DEX, Solarbio, Beijing, China), 0.5 mmol/L 3-isobutyl-1-methylxanthine (IBMX, Solarbio, Beijing, China), and 10 μg/mL insulin (Solarbio, Beijing, China). The medium was changed every 48 h. After 4 days of induction, the formation of lipid droplets in preadipocytes was observed. Subsequently, the medium was replaced with a maintenance medium containing 5% FBS and 10 μg/mL insulin for another 4 days. Finally, the medium was replaced with a growth medium (GM) to promote the maturation of adipocytes.

### 4.3. Plasmid Construction and Cell Transfection

The MSTRG4710-overexpressing (pcDNA3.1(+)-MSTRG4710) plasmids were constructed by cloning the entire length of MSTRG4710 gene coding region into pcDNA3.1 vectors (Beijing Tsingke Biotech Co., Ltd., China). The short interfering RNAs (siRNAs) of *IGF1* (si-IGF1) and *MSTRG4710* (siMSTRG4710), miR-29b-3p mimic (mimics), inhibitor were directly synthesized from Beijing Tsingke Biotech Co., Ltd. In the proliferation assay, Preadipocytes were cultured in 24-well or 6-well plates and transfected with the mimics, inhibitor, NC, and inhibitor-NC of miR-29b-3p, overexpressing plasmids, and siRNA, respectively, when cell density reached about 60% confluence. In the differentiation assay, transfection with the mimics described above was performed when the rabbit preadipocytes had proliferated to more than 80%. According to the manufacturer's instructions, the miR-29b-3p, overexpressing plasmids, and siRNA were transfected with Lipofectamine 3000 (Invitrogen, Carlsbad, CA, USA) at 50 and 100 nM/well concentrations for 6 h. Then, the transfection reagent was washed twice with PBS and added into DMEM/F12 containing 10% FBS culture for two days.

### 4.4. Rapid Amplification of cDNA Ends Assay

Following the manufacturer's specifications, the total RNA extracted from rabbit preadipocytes was used to rapidly amplify cDNA ends (RACE) PCR using the SMARTer RACE 5′/3′ Kit (Takara, Beijing, China). The primers utilized for 5′-RACE and 3′-RACE can be found in Appendix A. After separation via agarose gel electrophoresis, the products of 5′-RACE and 3′-RACE were Sanger sequenced when single and clean bands were obtained. The sequencing results were assembled using DNASTAR Lasergene v7.1 software to obtain the complete sequence.

### 4.5. Target Gene Prediction and Luciferase Reporter Assay

This study used the dual luciferase reporter system to detect the interaction between miRNA, mRNA and lncRNA, using miRwalk and miRBase to predict miR-29b-3p target genes based on sequence homology. The database (http://www.targetscan.org/vert_71/, accessed on 18 December 2021) was used to predict the hybridization of miR-29b-3p with the 3’ UTR of *IGF1*. We predicted the potential binding site between the miRNA and lncRNA using DNASTAR. Lasergene v7.1 based on the sequence information, respectively. Luciferase reporter plasmids (wild-type (WT) and mutant of target sequence) were constructed by Tsingke Biotechnology Co., Ltd. (Beijing, China). Next, 293T cells were seeded into 24-well plates (NEST Biotechnology, Wuxi, China). When the cell density reached 80%, miR-29b-3p mimics or NC were co-transfected with specific wild-type (WT) or mutant plasmids using Lipofectamine 3000 reagents. After 24 h, luciferase activities were measured using the Duo-Lite TM Luciferase Assay System from Vazyme (Nanjing, China).

### 4.6. Oil Red O Staining and Lipid Quantification

We inoculated preadipocytes into the 35 mm cell culture dishes (NEST Biotechnology, Wuxi, China). The cells for Oil Red O staining should be washed twice with PBS and fixed in 10% paraformaldehyde for 30 min. The newly purchased Oil Red O solution was diluted with DEPC water at a ratio of 3:2, filtered to obtain Oil Red O (NJBI, Nanjing, China) working solution, and used to stain the cells for 25 min. After staining, the cells were washed with PBS and imaged under an inverted microscope. Additionally, the number of lipid droplets was measured using image-pro plus 6.0 software (Media Cybernetics, Rockville, MD, USA).

### 4.7. Cell Proliferation Analysis CCK-8 and EDU Proliferation Assay

The fourth generation of rabbit preadipocytes was seeded in 96-well cell culture plates with six replicates per experimental group, and the volume of culture medium per well was 100 μL. When the cell reached about 70% confluency, we performed cell transfection. After transfection, the growth medium was changed to 6h. After culturing for 24h, add 10 μL of CCK-8 (Sangon Biotech, Shanghai, China) reagent to each well and incubate for 2 h. The absorbance values of all samples were measured at 450 nm using an automatic microplate reader. When the cells reach 60–70% confluence, transfected preadipocytes according to the above method. Then, the cells were cultured in GM for 24 h post-transfection and then incubated for 12 h in a medium containing 50 μM EDU (RiboBio, Guangzhou, China) before immunostaining. Subsequently, the cells were fixed, penetrated, and stained following the instructions provided with the EDU kit. However, the staining time may vary and should be adjusted based on cell density. At least 6 images per group were captured using a fluorescence microscope. The number of nuclei and EDU incorporation were analyzed using Image J software (V1.8.0, National Institutes of Health, Bethesda, MD, USA). The ratio of the number of nuclei between EDU incorporation represented the percentage of EDU-positive cells.

### 4.8. Quantitative Real-Time Polymerase Chain Reaction Analysis

According to the instructions, RNAiso reagent (Takara, Beijing, China) was used to extract total RNA from cells and tissues. The purity of total RNA was assessed using a NanoDrop 2000 UV-Vis spectrophotometer (Thermo, Waltham, MA, USA) by measuring the A260/A280 and A260/A230 ratios. Additionally, the integrity of the RNA was evaluated through 1.0% agarose gel electrophoresis. The first strand cDNA synthesis of total RNA and small RNA was performed using the prime PrimeScript RT Reagent Kit (Takara, Beijing, China) and SYBR^®^ PrimeScript™ miRNA Reverse Transcription Kit (Takara, Beijing, China) following the instructions provided with the respective kits. Subsequently, the corresponding cDNA was stored at −20 °C. For qPCR analysis, SYBR Green qPCR Master Mix (Best Enzyme, Shanghai, China) was used on a CFX96 system (Bio-Rad, Hercules, CA, USA). qPCR was performed using GAPDH and U6 as housekeeping genes in a reaction volume of 10 μL. The primer sequences are listed in Appendix A. The miR-29b-3p reverse primer and U6 primers were provided in the reagent kits. The 2^−ΔΔCt^ method was used to analyze the relative expression of each gene. qPCR was performed using GAPDH and U6 as housekeeping genes in a reaction volume of 10 μL. The primer sequences are listed in Appendix A. The miR-29b-3p reverse primer and U6 primers were provided in the reagent kits.

### 4.9. Western Blot Analysis

The total protein from the cells was collected using a commercial Protein Extraction Kit (Solarbio, Beijing, China), following the kit instructions. The protein concentration was measured using the Bradford protein assay kit (Novoprotein, Shanghai, China). Subsequently, the protein concentration was determined using the Bradford protein assay kit (Novoprotein, Shanghai, China), and only proteins meeting the criteria (concentration > 10 mg/mL) were used for further experimentation. Briefly, the protein (35–50 μg) was resolved on 4–12% SDS-PAGE (ebio-ACE, Changzhou, China) at 180 V for 30 min and transferred to a PVDF membrane at 400 mA for 25 min. The membrane was then blocked with skim milk powder for 2h. The membranes were incubated with the correspondingly primary antibodies at 4 °C for 8 h and subsequently incubated with the secondary antibodies for 2 h [Goat anti-rabbit IgG H&L (HRP), Zen bioscience, Chengdu, China (antibody: PBS = 1:10,000)]. Subsequently, the bands were washed three times. The membranes were subjected to an ECL chemiluminescence reagent (HAKATA, Shanghai, China) to detect immunoreactivities. The Touch Imager Pro, e-BLOT Life Science (Shanghai, China) Co., Ltd., captured images. The primary antibodies PCNA and CDK4 were purchased from Zen bioscience (Chengdu, China) (antibody: PBS = 1:1000). The primary antibodies PPARγ, C/EBPα, FABP4, SREBP1, and IGF1 were purchased from the Abclonal (Wuhan, China) (antibody: PBS = 1:1000). The primary antibodies β-actin purchased from the Abclonal (Wuhan, China) (antibody: PBS = 1:5000).

### 4.10. RNA-Seq

After reaching 80% growth density, preadipocytes were transfected with the method described in method 3.3. The mimics, NC, siMSTRG4710 and siMSTRG4710-NC, were transfected, and the differentiation medium was replaced after 6 h. After two days, trizol was adopted to extract total RNA for RNA-Seq. The RNA samples were prepared and underwent rigorous quality control measures. The purity of the sample was determined via NanoPhotometer^®^ (IMPLEN, Westlake Village, CA, USA), and RNA integrity and concentration were assessed using the RNA Nano 6000 Assay Kit of the Bioanalyzer 2100 system (Agilent Technologies, Santa Clara, CA, USA). After the total RNA samples were tested for qualification, the MGIEasy RNA library preparation kit was used for library construction. After purification, the double-stranded PCR library was unzipped and looped to form single-stranded circular DNA. The rolling circle amplification (RCA) technology creates a DNA nanoball (DNB). The DNB is loaded into the chip through the automatic sample loading system and fixed. The high-throughput sequencing was performed using the MGI DNBSEQ-T7 sequencing platform with a sequencing read length of PE150. Using Combinatorial Probe Synthesis, DNA molecular anchors and fluorescent probes are polymerized on DNA nanoballs, and optical signals are acquired by a high-resolution imaging system. High-quality and high-accuracy sample sequence information is obtained after the digital processing of optical signals. Differential expression analysis was performed using the DESeq2 (25 June 2023) with *p* ≤ 0.05.

### 4.11. Statistical Analysis

Statistical analysis was performed using GraphPad Prism v6.0 software (San Diego, CA, USA). They were consistent with normal distribution and presented as means ± SEM. Student’s *t*-tests were used to analyze the significance of differences between the groups. Moreover, multiple comparisons were performed using one-way or two-way ANOVA analysis. When * *p* < 0.05, the differences were significant. 

## 5. Conclusion

In summary, we identified MSTRG4710 as an essential factor associated with obesity, which plays a crucial role in adipocyte differentiation and adipogenesis. Our study is the first to demonstrate that knockdown of MSTRG4710 in rabbit preadipocytes significantly inhibits proliferation and differentiation, while overexpression of MSTRG4710 promotes these processes. Moreover, our study presents the novel discovery of the interaction between MSTRG4710 and miR-29b-3p. We provide evidence that MSTRG4710 functions as a regulator of *IGF1* expression by acting as a sponge for miR-29b-3p in preadipocytes. This research enriches the content of lncRNAs and ceRNAs in adipocyte development and provides new targets for treating obesity and other metabolic syndromes. Many factors regulate the development of adipose tissue. In addition to miRNA and lncRNAs, many signaling pathways are also involved, such as the Wnt/β-catenin and TGF-β signaling pathways. However, the impact of lncRNAMSTRG4710 on downstream signaling pathways associated with adipogenic differentiation remains unexplored. The sequencing results also found that miR-29b-3p and MSTRG4710 were involved in the related lipid metabolism pathways. However, the specific mechanism underlying their role remains unclear, necessitating further studies for a comprehensive understanding.

## Figures and Tables

**Figure 1 ijms-24-15715-f001:**
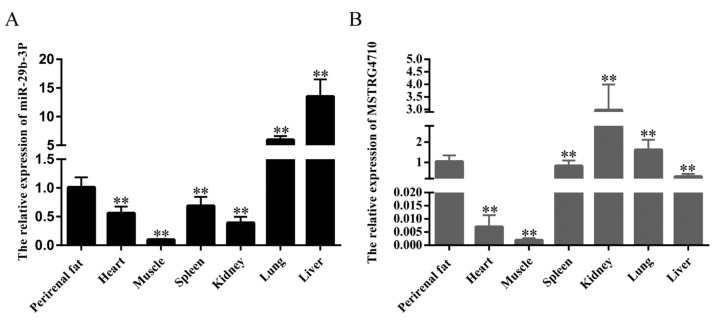
Tissue expression profiles of lncRNA-MSTRG4710 and miR-29b-3p in rabbits. (**A**) The relative expression of miR-29b-3p. (**B**) The relative expression of MSTRG4710. The data are presented as means ± SEM (*n* > 6). ** *p* < 0.01.

**Figure 2 ijms-24-15715-f002:**
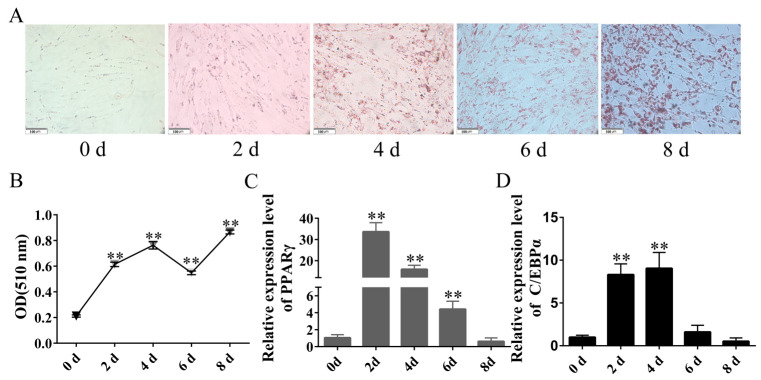
Isolation and culture and establishment of rabbit differentiation model of rabbit preadipocytes. (**A**) Oil Red O staining of lipid droplets at 0, 2, 4, 6, and 8 days of differentiation. (**B**) Quantitative detection of Oil Red O staining. (**C**) Relative expression levels of *PPARγ* mRNA. (**D**) Relative expression levels of *C/EBPα* mRNA. The data are presented as means ± SEM (*n* = 9). ** *p* < 0.01.

**Figure 3 ijms-24-15715-f003:**
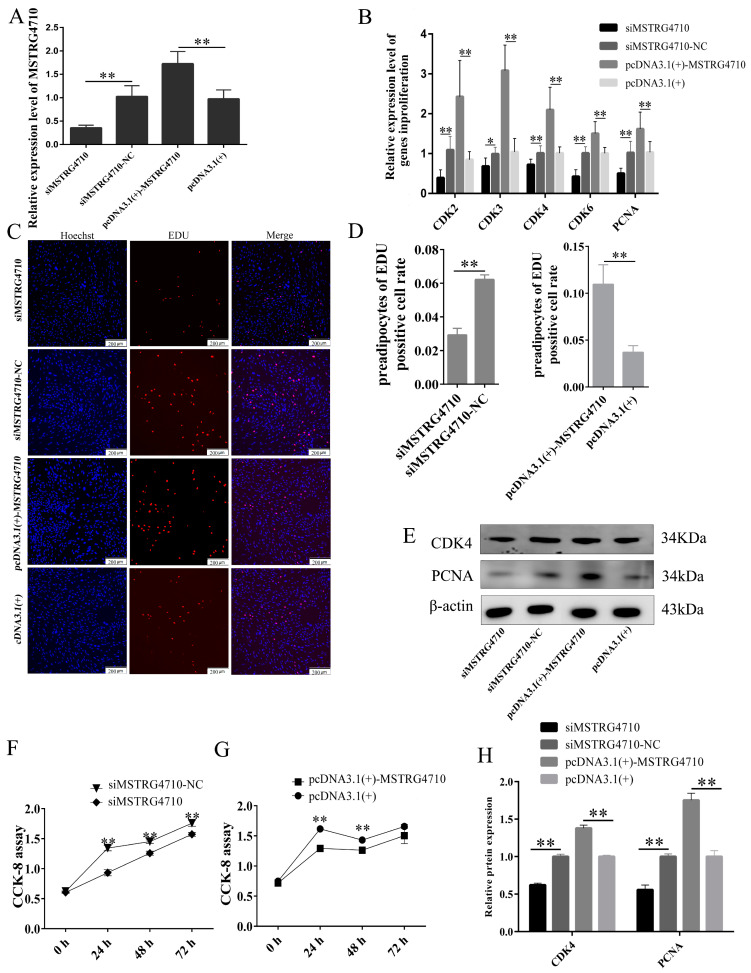
LncRNA-MSTRG4710 promotes the proliferation of rabbit preadipocytes. (**A**) The relative expression levels of MSTRG4710 in preadipocytes of rabbit-induced proliferation at 2 d after transfecting. (**B**) The relative expression levels of *CDK2*, *CDK3*, *CDK4*, *CDK6* and *PCNA* in preadipocytes of rabbit-induced proliferation at 2 d after transfecting. (**C**) The picture of the EDU proliferation assay for preadipocytes of rabbit-transfected. Red fluorescence represents the EDU-positive cells, and blue fluorescence represents the Hoechst-stained cells. (**D**) The percent of EDU-positive cells. EDU-positive cells rate = EDU-positive cells/Hoechst-stained cells × 100% (*n* = 3). (**E**,**H**) CDK4 and PCNA protein levels. (**F**,**G**) The absorbance of preadipocytes of rabbit at 0, 2, 4 and 6 d after transfecting with the siMSTRG4710, siMSTRG4710-NC, pcDNA3.1(+)-MSTRG4710, and pcDNA3.1(+) (*n* = 6). The data are presented as means ± SEM. * *p* < 0.05; ** *p* < 0.01.

**Figure 4 ijms-24-15715-f004:**
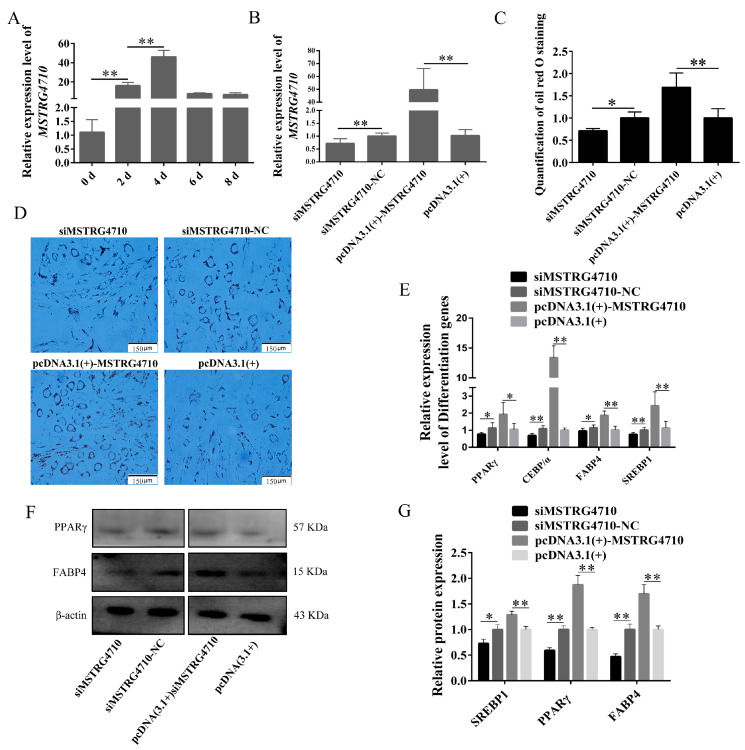
LncRNA-MSTRG4710 promotes the differentiation of rabbit preadipocytes. (**A**) The relative expression levels of MSTRG4710 in preadipocytes of rabbit-induced differentiation at 0 d, 2 d, 4 d, 6 d, 8 d. (**B**) The relative expression levels of MSTRG4710 in preadipocytes of rabbit-induced differentiation at 2 d after transfecting with the siMSTRG4710, siMSTRG4710-NC, pcDNA3.1(+)-MSTRG4710, and pcDNA3.1(+). (**C**) The number of lipid droplets (*n* = 3) after 2 d of transfection. (**D**) Oil Red O staining of lipid droplets after 2 d of transfection. (**E**) The relative expression levels of *PPARγ*, *C/EBPα*, *FABP4* and *SREBP1* in preadipocytes of induced differentiation at 2 d after transfecting. (**F**,**G**) FABP4, PPARγ and SREBP1 protein levels. The data are presented as means ± SEM. * *p* < 0.05; ** *p* < 0.01.

**Figure 5 ijms-24-15715-f005:**
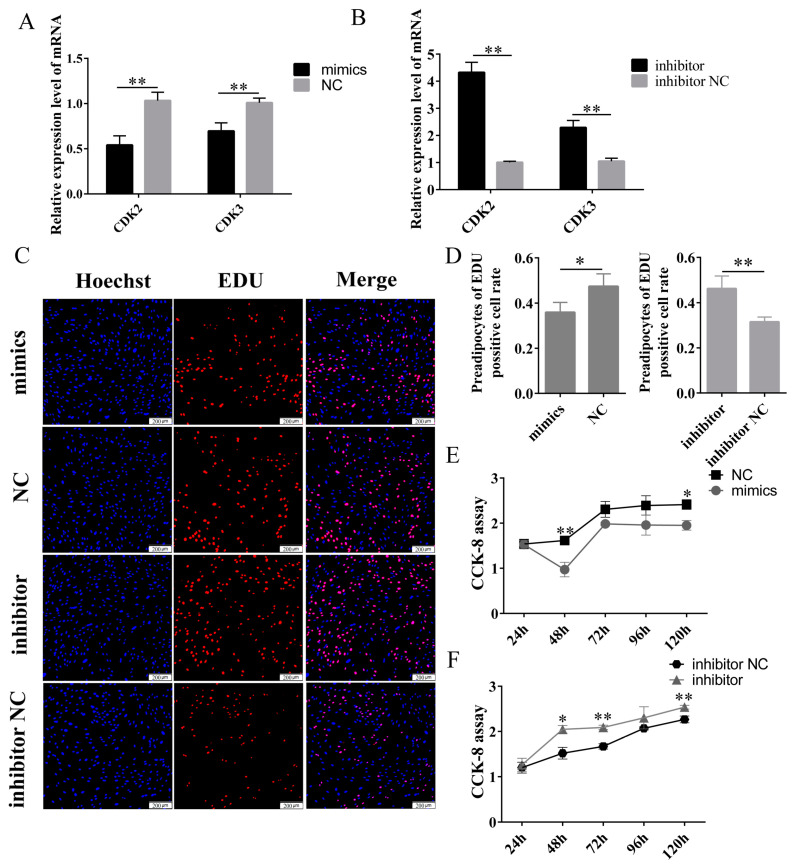
miR-29b-3p inhibited the proliferation of rabbit preadipocytes. (**A**,**B**) The relative expression levels of *CDK2* and *CDK3* in preadipocytes of rabbit induced proliferation at 2 d after transfecting with mimics, NC, inhibitor, inhibitor N.C. (**C**) The picture of the EDU proliferation assay for preadipocytes of rabbit-transfected. Red fluorescence represents the EDU-positive cells, and blue fluorescence represents the Hoechst-stained cells. (**D**) The percent of EDU-positive cells. EDU-positive cells rate = EDU-positive cells/Hoechst-stained cells × 100% (*n* = 3). (**E**,**F**) The absorbance of preadipocytes of rabbit at 0, 2, 4 and 6 d after transfecting with the mimics, NC, inhibitor, inhibitor N.C. The data are presented as means ± SEM. * *p* < 0.05; ** *p* < 0.01.

**Figure 6 ijms-24-15715-f006:**
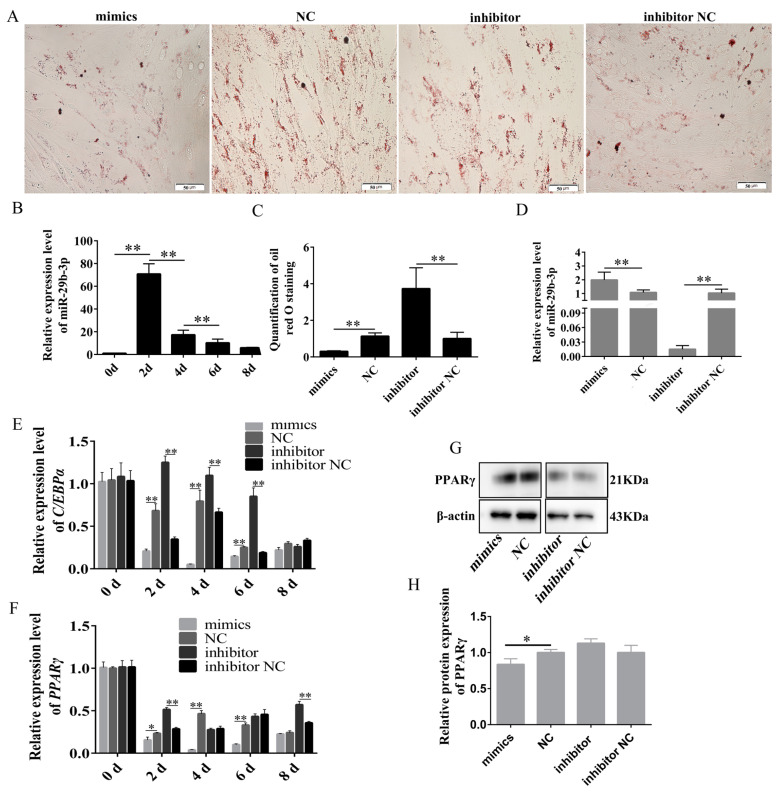
miR-29b-3p inhibited the differentiation of rabbit preadipocytes. (**A**) Oil Red O staining of lipid droplets after 2 d of transfection. (**B**) The relative expression levels of miR-29b-3p in preadipocytes of rabbit-induced differentiation at 0 d, 2 d, 4 d, 6 d and 8 d. (**C**) The number of lipid droplets after 2 d of transfection (*n* = 3). (**D**) The relative expression levels of miR-29b-3p in preadipocytes of rabbit-induced differentiation at 2 d after transfecting with the mimics, NC, inhibitor, inhibitor NC. (**E**,**F**) The relative expression levels of *PPARγ*, *C/EBPα* in preadipocytes of induced differentiation at 0 d, 2 d, 4 d, 6 d and 8 d after transfecting. (**G**,**H**) PPARγ protein levels. The data are presented as means ± SEM. * *p* < 0.05; ** *p* < 0.01.

**Figure 7 ijms-24-15715-f007:**
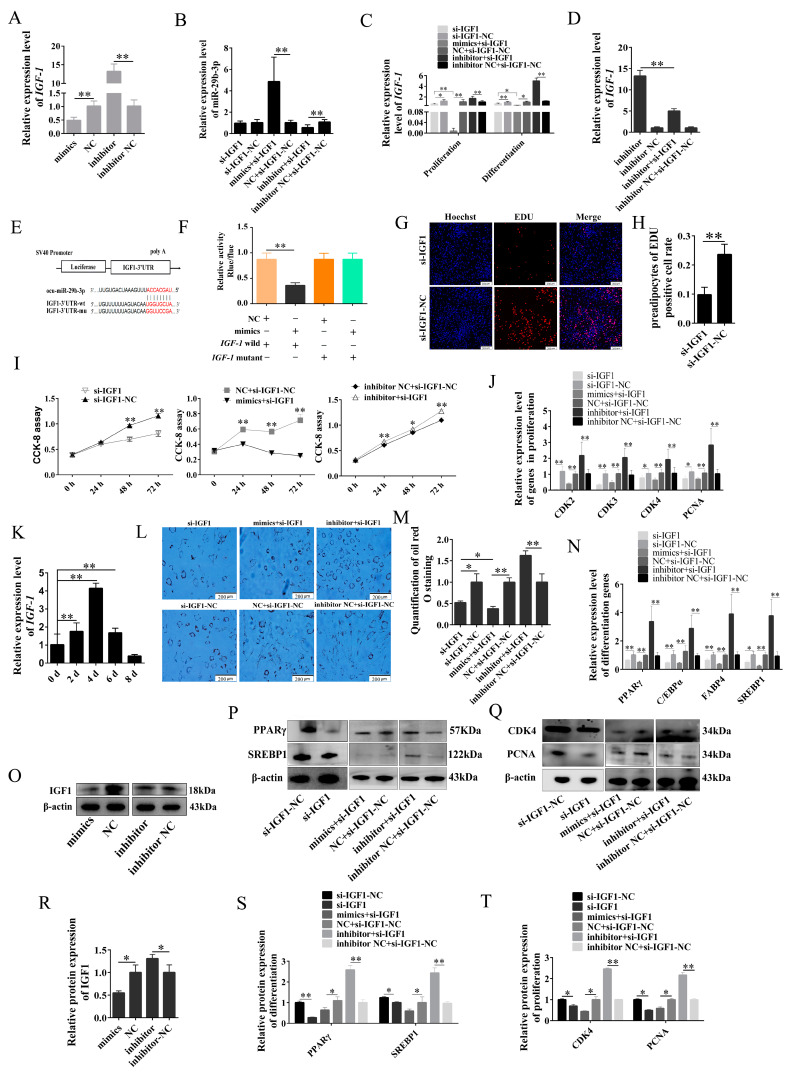
miR-29b-3p inhibits the proliferation and differentiation of rabbit preadipocytes by targeting *IGF1*. (**A**) The relative expression levels of IGF1 were measured two days after differentiation of preadipocytes transfected with mimic, NC, inhibitor, and inhibitor N.C. (**B**) The relative expression levels of miR-29b-3p in preadipocytes of rabbit-induced differentiation at 2 d after transfecting with si-IGF1, si-IGF1-NC, mimics+ si-IGF1, NC + si-IGF1-NC, inhibitor+ si-IGF1 and inhibitor NC+ si-IGF1-NC. (**C**) The relative expression levels of *IGF1* in preadipocytes of rabbit-induced differentiation and proliferation at 2 d after transfecting. (**D**) The relative expression levels of *IGF1* in preadipocytes of rabbit-induced proliferation at 2 d after transfecting with inhibitor, inhibitor NC, inhibitor + si-IGF1 and inhibitor NC + si-IGF1-NC. (**E**) The predicted binding site of gene *IGF1* with miR-29b-3p. (**F**) Luciferase assays were performed via co-transfection of *IGF1* WT and mutant plasmids with miR-29b-3p mimic and NC, respectively, in 293T cells, and the WT + NC group was used as the control group (*n* = 3). (**G**) The picture of the EDU proliferation assay for preadipocytes of rabbit-transfected. Red fluorescence represents the EDU-positive cells, and blue fluorescence represents the Hoechst-stained cells. (**H**) The percent of EDU-positive cells. EDU-positive cells rate. (**I**) The absorbance of preadipocytes of rabbit at 0, 2, 4 and 6 d after transfecting. (**J**) The relative expression levels of *CDK2*, *CDK3*, *CDK4* and *PCNA*. (**K**) The relative expression levels of *IGF1* during preadipogenic differentiation. (**L**) Oil Red O staining of lipid droplets after 2 d of transfection. (**M**) The number of lipid droplets (*n* = 3) after 2 d of transfection. (**N**) The relative expression levels of *PPARγ*, *C/EBPα*, *FABP4* and *SREBP1*. (**O**–**T**) The IGF1, CDK4, PCNA, FABP4, PPARγ and SREBP1 protein levels. The data are presented as means ± SEM. * *p* < 0.05; ** *p* < 0.01.

**Figure 8 ijms-24-15715-f008:**
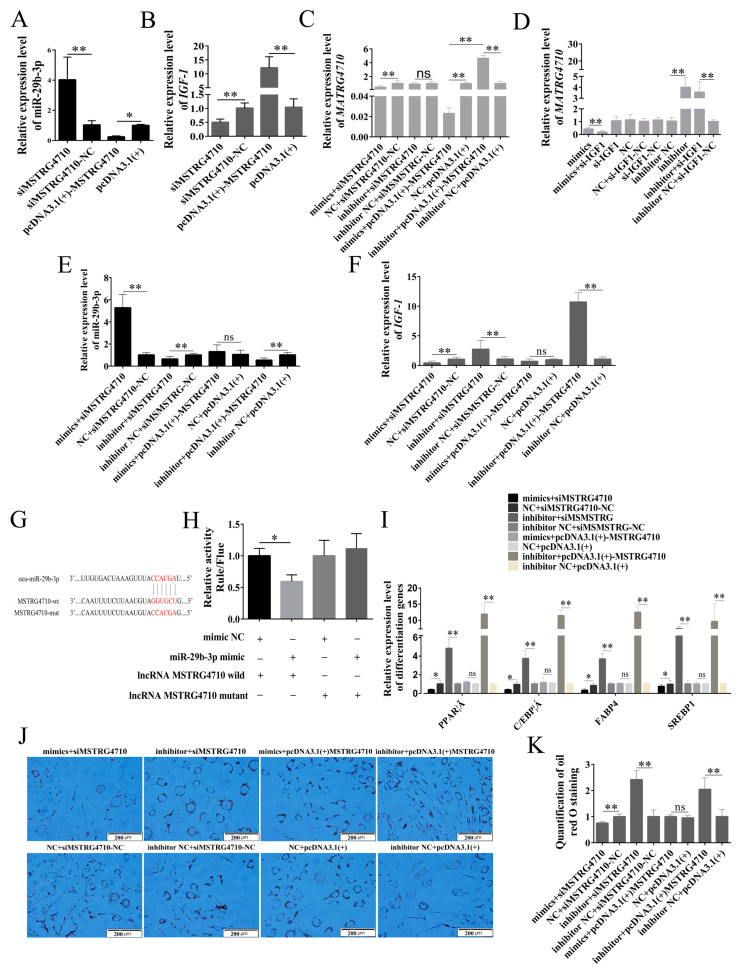
MSTRG4710 promotes the differentiation of rabbit preadipocytes via the miR-29b-3p/IGF1 pathway. (**A**,**E**) The relative expression levels of miR-29b-3p in rabbit preadipocytes induced differentiation at 2 d after transfecting. (**C**,**D**) The relative expression levels of MSTRG4710 in preadipocytes of rabbit-induced differentiation at 2 d after transfecting. (**B**,**F**) The relative expression levels of *IGF1* in preadipocytes of rabbit-induced differentiation. (**G**) The predicted binding site of gene MSTRG4710 with miR-29b-3p. (**H**) Luciferase assays were performed via co-transfection of MSTRG4710 WT and mutant plasmids with miR-29b-3p mimic and NC, respectively, in 293T cells, and the WT + NC group was used as the control group (*n* = 3). (**I**) The relative expression levels of *PPARγ*, *C/EBPα*, *FABP4* and *SREBP1*. (**J**) Oil Red O staining of lipid droplets after 2 d of transfection. (**K**) The number of lipid droplets (*n* = 3) after 2 d of transfection. The data are presented as means ± SEM. * *p* < 0.05; ** *p* < 0.01; ns = no significant.

**Figure 9 ijms-24-15715-f009:**
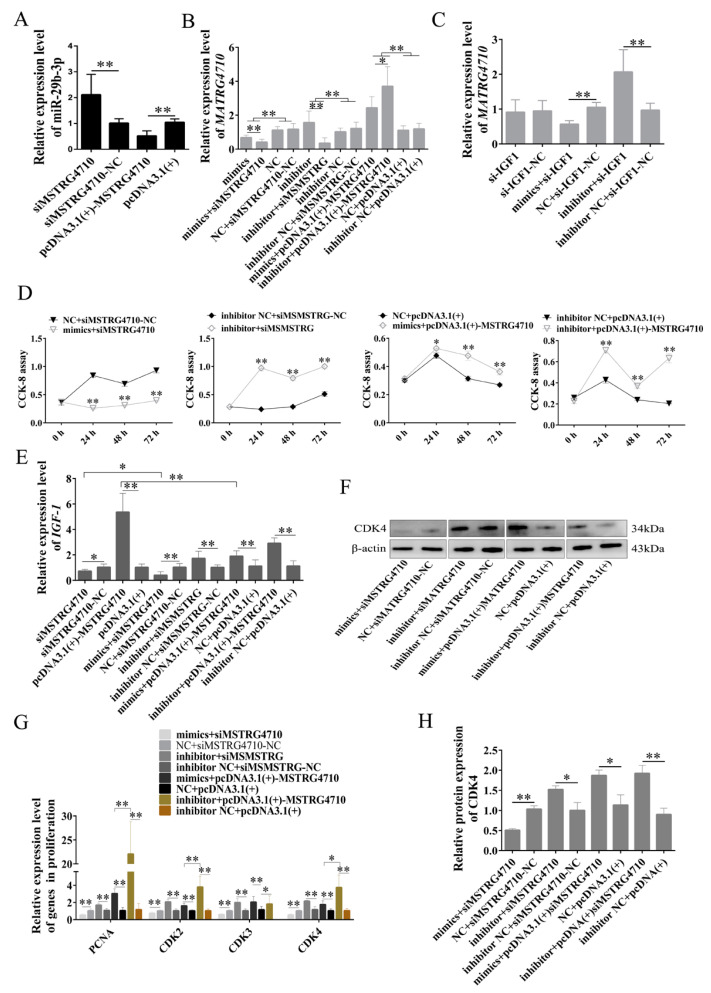
MSTRG4710 promotes the proliferation of rabbit preadipocytes via the miR-29b-3p/IGF1 pathway. (**A**) The relative expression levels of miR-29b-3p in preadipocytes of rabbit-induced proliferation at 2 d after transfecting. (**B**,**C**) The relative expression levels of MSTRG4710 in preadipocytes of rabbit-induced proliferation at 2 d after transfecting. (**D**) The absorbance of preadipocytes of rabbit at 0, 2, 4 and 6 d after transfecting. (**E**) The relative expression levels of *IGF1* in preadipocytes of rabbit-induced proliferation at 2 d after transfecting. (**F**,**H**) The CDK4 protein levels. (**G**) The relative expression levels of *PCNA*, *CDK2*, *CDK3* and *CDK4*. The data are presented as means ± SEM. * *p* < 0.05; ** *p* < 0.01.

**Figure 10 ijms-24-15715-f010:**
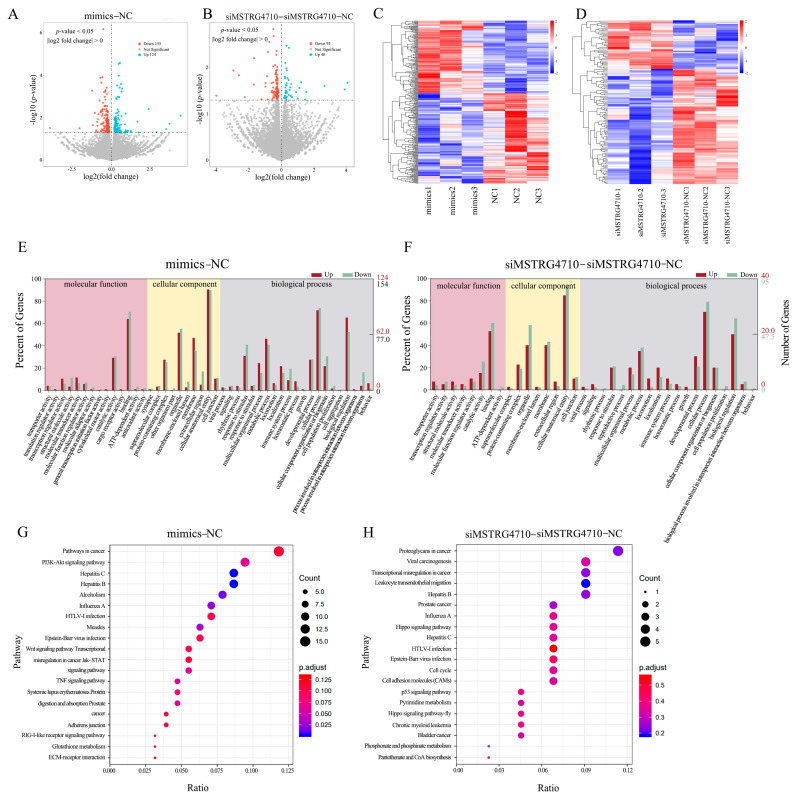
Analysis of differentially expressed genes. (**A**,**B**) The volcano plot was constructed for the differentially expressed genes based on log2 (fold change) and −log10 (*p*-value). (**C**,**D**) Heat map of differentially expressed genes in each comparison group. According to the results of differential expression analysis, *p* < 0.05 was extracted log2 (fold change) > 0 genes, and then a heat map was made based on the expression of each sample. The blue, white and red gradient indicates the expression level of the gene, with blue indicating a lower expression level and red indicating a higher expression level. (**E**,**F**) GO analysis of the differentially expressed genes was performed, imaging only those terms significantly enriched in the BP, CC, and MF categories. The abscissa is the second-level GO entry of the annotation results of differentially expressed genes, the left ordinate is the proportion of upregulated/downregulated differentially expressed genes, and the right ordinate is the number of upregulated/downregulated differentially expressed genes. (**G**,**H**) KEGG analysis of the differentially expressed genes only listed significant enrichment pathways.

**Table 1 ijms-24-15715-t001:** mimics, NC, siMSTRG4710 and siMSTRG4710-NC groups data output quality statistics.

Sample	Clean Reads	Q30 (%)	GC (%)	Total Mapping (%)	Exon (%)	Intron (%)	Intergenic (%)
mimics 1	46,291,190	94.67	56.41	87.80	67.35	15.37	17.28
mimics 2	67,471,608	95.06	56.73	87.88	68.92	14.23	16.85
mimics 3	44,188,296	94.69	56.71	87.74	67.52	15.48	17.00
NC1	57,343,036	94.83	56.15	88.39	67.38	15.72	16.89
NC2	46,680,152	94.12	56.70	87.70	68.69	15.40	15.91
NC3	58,249,758	94.89	56.64	88.07	67.69	15.08	17.22
siMSTRG4710-1	75,961,000	95.25	56.64	88.43	65.86	16.63	17.51
siMSTRG4710-2	75,068,706	95.47	57.32	87.75	66.61	16.80	16.59
siMSTRG4710-3	46,202,572	94.50	56.96	87.20	67.67	14.71	17.62
siMSTRG4710-NC1	46,667,670	94.33	56.50	87.85	66.54	16.60	16.86
siMSTRG4710-NC2	44,768,540	94.78	56.78	87.89	66.95	16.78	16.27
siMSTRG4710-NC3	74,122,238	95.43	56.85	88.02	67.69	15.45	16.86

## Data Availability

All data generated or analyzed during this study are included in this published article.

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
