# Peer review of "lncRNA MSTRG4710 Promotes the Proliferation and Differentiation of Preadipocytes through miR-29b-3p/IGF1 Axis"

_ijms, 2023, doi:10.3390/ijms242115715_

Round 1
Reviewer 1 Report
1- Several data are unclear
Fig. 3C: the overexpression upon transfection is rather poor (also compared with Fig. 4B) – pls. comment
Fig. 3H: why are no differences visible in the CDK4 bands? (compare with Fig. 3E)
Fig. 3G, I: effects only occur in the first 24 h, then lines essentially run in parallel. Please comment and correct in the text.
Fig. 4B: poor knockdown efficacy – why?
Fig. 4D: why are the 2nd and 3rd bars similarly high, but not the negative controls (2nd and 4th bar)?
Fig. 5C: Hoechst staining is too dark. Why is the EDU staining different between the two negative controls, while it is similar between NC and inhibitor?
Fig. 5D: again, why are the 2nd and 3rd bars similar, but not the negative controls?
Fig. 6A: why down the negative controls major differences, while NC and inhibitor look similar? The same applies to Fig. 6C and Fig. 6G.
Fig. 7A: inconsistent results between d4, d6, d8
Fig. 7J: again, why are the two negative controls different in the IGF1 band?
Line 405: ‘… miR-29b-3p was significantly increased in the si-IGF1 group.’ – why?
Line 508: why ‘miR-383-5p’?
2- Wordings and statements need correction or clarification
Line 235-236: ‘… were also highly expressed in perirenal adipose tissues …’ – Fig. 1A does not support this conclusion
Line 172-174: “The adipocytes…” – unclear sentence
Line 278: ‘… was extremely significantly increased…’ – no.
Line 396 – 408: the figures are not described in the text according to their order in the composite figure
3- Some Figures need improvement, several figures and esp. figure legends are too small
Fig. 2A, B: the background is different which does not allow direct comparison of the different time points.
Fig. 2D: why is the time axis different from Fig. 2C and Fig. 2E?
Fig. 3A: too small, rather transfer into the Supplement
Fig. 3B: not required
Fig. 3G, I: here and elsewhere (e.g., Fig. 5E, F), symbols are too small to distinguish between the different lines
Fig. 3G-I: wrong order in the composite figure – needs re-arrangement
Fig. 4F: poor quality
Fig. 7I: labeling and symbols too small
Fig. 8: many labelings are too small
4- The paper would benefit from including a scheme on the miR-29b-3p/IGF1 axis
5- The Methods section needs improvement in wording and the description of some methods – please review and correct thoroughly
6- What is a ‘manufacturer’s agreement’? What means ‘forebody’ in the context of this paper?
7- Some items can be transferred to the Supplement: Table 1, Table 2
8- Several typos throughout the paper need correction
see above
Author Response
Dear Reviewer:
Many thanks to the reviewer for taking time out of his busy schedule to review this manuscript. Thank you for your valuable comments. Comments are of great help in improving my article. We have also responded to your comments one by one, and the specific content of the reply is attached. I hope you can spare time to check the attachment in your busy schedule.

Reviewer 2 Report
Overall, this is an interesting study that shows that lncRNA MSTRG4710 may be involved in controlling the proliferation and differentiation of preadipocytes. Nevertheless, I have some comments for the Authors.
- Please provide the gender of the animals used to isolate cells. Furthermore, the Authors should justify why they used rabbits as donors of adipose tissue samples.
- Sequences of primers. Gene accession no. should be added to the Tables.
- Western blot should be described in a more detailed fashion. For example, how many micrograms of proteins were loaded on the gel? Furthermore, dilutions and cat. no of antibodies should be added. This is important to allow others to reproduce data.
- Statistical analysis. Can Authors explain why they used a t-test? This kind of test can be used to compare only two groups only. In this paper, in some experiments, more
- Fig. 2A-B, I cannot read scale bare. The same problem is percent in all pictures in the manuscript.
- The material and methods should provide more details regarding RNAseq (section 2.10) and data analysis.
- Overall, some graphs should be better described. While Y-charts have titles, units are missing.
- Overall, the discussion is well-written. Nevertheless, it can be improved by listing and discussing the limitations of this work.
- The no. Of ethical approval is missing.
Author Response

(The authors gave the same response as above.)

Round 2
Reviewer 2 Report
I have no more comments for the Authors.